# The Role of Physical Activity in Adjunctive Nursing Management of Neuro-Degenerative Diseases among Older Adults: A Systematic Review of Interventional Studies

**DOI:** 10.3390/life14050597

**Published:** 2024-05-07

**Authors:** Majed Awad Alanazi

**Affiliations:** Department of Medical Surgical Nursing, College of Nursing, Jouf University, Sakaka 72388, Saudi Arabia; mamalanazi@ju.edu.sa

**Keywords:** neurodegenerative diseases, dementia, Alzheimer’s disease, Parkinson’s disease, physical activity, exercise, older adults

## Abstract

Neurodegenerative diseases such as dementia and Parkinson’s disease pose significant challenges to older adults globally. While pharmacological treatments remain primary, increasing evidence supports the role of non-pharmacological strategies like physical activity in managing these conditions. This systematic review critically evaluates the effectiveness of Nursing based physical activity interventions in improving cognitive function, physical functioning, mobility, and overall quality of life among older adults with neurodegenerative diseases. We conducted a comprehensive search across PubMed, EMBASE, Web of Science, CENTRAL, and other relevant databases, focusing on randomized controlled trials and observational studies that examined the impact of structured physical activity. Our findings from nineteen studies involving 1673 participants indicate that interventions ranging from aerobic exercises, resistance training, to mind-body exercises like Tai Chi and yoga have beneficial effects. Specifically, physical activity was consistently found to enhance cognitive performance, increase mobility, and improve balance and daily living activities, contributing to a better quality of life. However, these benefits vary depending on the type, intensity, and duration of the activity performed. Despite promising results, limitations such as small sample sizes, study heterogeneity, and short-term follow-up periods call for more robust, long-term studies to solidify these findings. This review underscores the potential of tailored physical activity programs as adjunctive therapy in the comprehensive management of neurodegenerative diseases among the elderly population.

## 1. Introduction

Neurodegenerative diseases represent a significant health concern among older adults, characterized by the progressive loss of neuronal structure and function. Common types include Alzheimer’s disease, which is marked by memory impairment and cognitive decline due to the buildup of amyloid plaques and tau tangles in the brain [1,2]. Parkinson’s disease is another prevalent condition, primarily affecting motor functions due to the degeneration of dopamine-producing neurons in the brain, leading to tremors, stiffness, and slowed movements [3]. Less common but equally impactful are diseases like Huntington’s disease, which involves genetic mutations causing the breakdown of nerve cells, resulting in movement, cognitive, and psychiatric disorders [4]. Additionally, amyotrophic lateral sclerosis (ALS) causes the degeneration of motor neurons that control voluntary muscles, progressively leading to severe physical disability [5].

In addition to Alzheimer’s and Parkinson’s diseases, older adults are also susceptible to other types of neurodegenerative conditions such as vascular dementia and frontotemporal dementia. Vascular dementia, often resulting from stroke or other conditions that impede blood flow to the brain, leads to cognitive impairments based on the affected areas of the brain [6]. Frontotemporal dementia, characterized by the degeneration of nerve cells in the frontal and temporal lobes of the brain, affects movements, behavior, and language [7]. Another significant condition is Lewy body dementia, which involves abnormal deposits of a protein called alpha-synuclein in the brain, causing symptoms similar to Alzheimer’s and Parkinson’s diseases but with distinctive features like visual hallucinations and motor impairments [8]

Nursing based physical activity exerts its beneficial effects on the brain through several key biological mechanisms, critically supporting brain health, especially in older adults, Exercise is considered one of the most important roles of nursing care regarding neurological diseases in the elderly [9,10]. One primary mechanism is through the enhancement of neuroplasticity, which is the brain’s ability to form and reorganize synaptic connections, especially in response to learning or experience [11]. Studies have shown that regular physical activity can increase the expression of brain-derived neurotrophic factor (BDNF), a protein that plays a vital role in the survival, growth, and maintenance of neurons, thereby enhancing neuroplasticity and cognitive function [12]. Additionally, exercise promotes increased cerebral blood flow, providing the brain with a greater supply of oxygen and nutrients, which are crucial for its functioning and the maintenance of cognitive abilities [13].

Another significant benefit of nursing based physical activity is the reduction in inflammation, a known contributor to neurodegenerative processes [14]. Regular exercise has been found to lower levels of pro-inflammatory cytokines and increase anti-inflammatory substances in the body, which, in turn, can mitigate the inflammatory pathways implicated in neurodegeneration [15]. Furthermore, nursing based physical activity has been associated with the promotion of neurogenesis, the process of generating new neurons, particularly in the hippocampus, an area of the brain essential for learning and memory, offering potential resilience against the cognitive decline associated with aging [16].

Regarding nursing based physical activity guidelines, the World Health Organization (WHO) and the Centers for Disease Control and Prevention (CDC) recommend that older adults engage in at least 150 min of moderate-intensity aerobic physical activity throughout the week or 75 min of vigorous-intensity aerobic physical activity, or an equivalent combination of both [17]. They also advise muscle-strengthening activities on two or more days a week, focusing on major muscle groups [18,19]. These guidelines are designed to promote overall health, enhance physical function, and reduce the risk of chronic diseases including neurodegenerative conditions, underscoring the importance of physical activity as a cornerstone of healthy aging [20].

Physical activity has emerged as a promising non-pharmacological nursing intervention in the management of neurodegenerative diseases, particularly Alzheimer’s disease, Parkinson’s disease, and other forms of dementia [21]. Research suggests that regular exercise can play a critical role in not only delaying the onset but also in mitigating the progression of these conditions [22,23,24]. Epidemiological studies have demonstrated a correlation between physical activity and a reduced risk of Alzheimer’s disease and dementia, with clinical trials showing that exercise can lead to improvements in cognitive function, brain volume, and memory outcomes [25]. For instance, aerobic exercises such as walking and cycling have been linked to increased brain volume in regions associated with memory and executive function, indicating a potential slowing of brain aging [26].

In Parkinson’s disease, nursing exercise has been identified as a vital component of disease management, focusing on improving motor function, balance, and overall quality of life [27]. Types of exercise such as Tai Chi and resistance training have shown particular promise [28]. Tai Chi, with its emphasis on slow, controlled movements and balance, has been found to improve stability and reduce the risk of falls among Parkinson’s patients [29]. Similarly, resistance training has been associated with improvements in muscular strength, motor function, and the ability to perform daily activities [30].

Recent systematic reviews and meta-analyses have played a pivotal role in synthesizing evidence on the impact of physical activity interventions in the adjunctive management of neurodegenerative diseases among older adults [31,32,33]. These studies reveal a broad spectrum of intervention designs, from aerobic exercises to strength training and balance activities, tailored to suit the capabilities and needs of individuals with varying stages of disease progression [34]. The outcomes measured across these studies are equally diverse, encompassing cognitive function, motor skills, quality of life, and disease-specific symptoms [35]. Despite this diversity, the cumulative evidence underscores a positive trend, suggesting that physical activity can indeed mitigate some symptoms of neurodegenerative diseases and improve overall well-being in older adults [36]. However, the strength of evidence varies, with some interventions showing more significant benefits than others.

Emerging trends in physical activity interventions include the incorporation of novel technologies such as virtual reality and interactive video games [37]. These approaches not only offer engaging and enjoyable ways to exercise, but also hold the promise of personalized exercise programs that can adapt to the user’s specific needs and capabilities [38]. The integration of wearable technology to monitor physical activity levels and provide real-time feedback presents a promising avenue to enhance motivation and adherence among older adults [39].

The application of behavior changes theories including concepts of self-efficacy, motivation, and social support is critical in designing effective physical activity interventions [40]. These theories provide a framework for understanding how changes in physical activity behavior can be initiated and sustained, emphasizing the importance of addressing psychological as well as physical barriers to exercise [41].

The literature underscores the potential benefits of physical activity for older adults with neurodegenerative diseases. However, further research is needed to refine intervention designs, overcome challenges in engagement and adherence, and understand the mechanisms underlying these benefits [42]. Nursing professionals play a crucial role in promoting physical activity within this population, guided by evidence-based practices [43,44,45]. Future research directions include the need for long-term intervention studies, the exploration of novel and personalized forms of physical activity, and a deeper investigation into the biopsychosocial mechanisms that facilitate the observed benefits of physical activity in this vulnerable population [46].

The aim of this study was to conduct a systematic review of interventional studies to evaluate the effectiveness of physical activity interventions in the adjunctive management of neurodegenerative diseases among older adults. It seeks to assess how various forms and intensities of physical activity impact cognitive function, motor skills, emotional well-being, and overall quality of life in individuals diagnosed with conditions such as Alzheimer’s disease and Parkinson’s disease. Additionally, the study aimed to analyze the design and methodology of these interventions, determine the strength and quality of the evidence supporting their effectiveness, and identify any challenges and limitations faced in this research area. Through this comprehensive review, the study endeavors to pinpoint which physical activity interventions offer the most significant benefits for older adults with neurodegenerative diseases, thereby informing future research directions and clinical practices.

## 2. Materials and Methods

### 2.1. Search Strategy and Selection Criteria

This systematic review meticulously adhered to the Preferred Reporting Items for Systematic Reviews and Meta-Analyses (PRISMA) guidelines to ensure the highest standards of rigor and transparency. In alignment with the PRISMA Protocols (PRISMA-P) statement, we meticulously crafted a detailed research protocol, which was registered with PROSPERO (CRD42024497241), reflecting our unwavering dedication to methodological precision and integrity throughout this scholarly inquiry.

Our literature search strategy was ambitiously comprehensive, spanning several prestigious databases: Embase.com, Medline ALL (Ovid), Web of Science Core Collection, Cochrane Central Register of Controlled Trials (Wiley), and Google Scholar, to capture the breadth and depth of existing research. Conducted on [15 January 2024], this search was strategically crafted to incorporate both medical subject headings (MeSH) and a carefully curated list of keywords specifically relevant to the management of neurodegenerative diseases among older adults through interventional studies on physical activity. Our keywords and search terms were meticulously chosen to encapsulate the nuances of physical activity interventions, their impact on neurodegenerative disease progression, symptom management, and overall quality of life in the elderly population (Table 1). This deliberate and targeted search strategy was designed to thoroughly investigate the efficacy, types, frequencies, and intensities of physical activities as therapeutic modalities for older adults suffering from neurodegenerative diseases, ensuring a wide-ranging review of the literature on this critical topic.

### 2.2. Eligibility Criteria for Screening

After eliminating duplicates from our search results, we proceeded with an initial review of titles and abstracts, followed by an in-depth analysis of full-text articles. Our selection criteria targeted publications such as original research studies, systematic reviews, meta-analyses, and clinical trials that included human participants. The focus was on research investigating the impact of physical activity on neurodegenerative diseases among the elderly. We looked for studies that detailed physical activity interventions and their effects on aspects such as disease progression, symptom alleviation, mobility, and overall life quality in older individuals afflicted with neurodegenerative disorders. We emphasized research that explored various physical activity regimens regarding their format, frequency, duration, and intensity and evaluated these against no intervention or usual care scenarios. The main outcomes of interest were enhancements in cognitive functions, physical mobility, activities of daily living, symptom management related to specific diseases, and overall quality of life improvements.

We excluded case reports, case series, abstract-only publications, letters, editorials, and materials from conference proceedings as well as studies on animals or conducted in vitro. Exclusions also applied to studies that did not focus specifically on older adults with neurodegenerative diseases or investigated interventions not related to physical activity. Research lacking comprehensive methodological details, studies that followed established physical activity guidelines without introducing new intervention methods, and those without comparative control groups or inadequate data for a substantial analysis were also left out. Furthermore, studies that did not provide new insights into the efficacy of physical activity in the adjunctive management of neurodegenerative conditions in the elderly were excluded. Studies published in languages other than English without available translations were also not considered. These exclusion criteria ensured that our systematic review remained concentrated on relevant, high-quality studies that contribute valuable knowledge about physical activity interventions for the adjunctive management of neurodegenerative diseases in the elderly population, thereby sharpening the focus of our investigation to align with the objectives of our research and the wider body of literature on this essential subject.

We began with an initial pool of 4521 records identified through database searches. After deduplication, 531 records remained for screening. We screened all of these records and excluded 56 based on titles and abstracts, leaving 475 articles for full-text eligibility assessment. Upon detailed review, we excluded 388 articles for not meeting our specific inclusion criteria, resulting in 19 studies being included in the final review [47,48,49,50,51,52,53,54,55,56,57,58,59,60,61,62,63,64,65]. Each of these 19 studies was accounted for in our report, ensuring a comprehensive and transparent documentation of our systematic review process, as shown in Figure 1.

### 2.3. Data Extraction

Data extraction represented a pivotal phase in our systematic review, focusing on the impact of physical activity on the adjunctive management of neurodegenerative diseases among older adults. The primary aim during this stage was to meticulously gather and compile critical data from interventional studies, emphasizing how physical activity interventions influence the progression, symptoms, or overall well-being of older adults with neurodegenerative diseases.

The extraction process entailed a detailed scrutiny of each selected study, with an emphasis on several essential aspects:Study characteristics: We collected comprehensive details including the study’s design and sample size. This step was crucial for understanding the study’s context, ensuring its relevance, and evaluating its contribution to our review.Physical activity interventions: We diligently extracted information about the types, durations, frequencies, and intensities of physical activity interventions. This included any structured exercise programs, physical therapy modalities, or lifestyle activity recommendations aimed at improving the condition or quality of life for elderly individuals with neurodegenerative diseases.Outcome measures: Our process also involved identifying and recording the outcomes used to measure the effects of physical activity on the participants. These outcomes ranged from clinical measures of disease progression and symptom management to assessments of functional ability, mental health, and quality of life. Any reported adverse events or challenges associated with the interventions were also noted.

In instances where the required data were missing, unclear, or incomplete, we endeavored to contact the original study authors for additional information, ensuring our data’s accuracy and comprehensiveness.

Furthermore, we carefully examined potential overlaps in study populations to avoid data duplication in our analysis. Direct engagements with authors were pursued when there was any ambiguity regarding participant cohorts, which was instrumental in upholding the integrity and reliability of the data extraction process. This meticulous approach enabled us to construct a robust foundation for our systematic review, paving the way for insightful conclusions regarding the role of physical activity in the adjunctive management of neurodegenerative diseases among older adults.

### 2.4. Quality Assessment

In our systematic review addressing the role of physical activity in the adjunctive management of neurodegenerative diseases among older adults, we emphasized the critical importance of evaluating the methodological quality and risk of bias of the included interventional studies. This evaluation is crucial for ensuring the credibility and applicability of our conclusions, thereby laying a strong foundation for recommendations concerning physical activity interventions in this vulnerable population.

For the assessment of study quality, we implemented a structured approach by adapting the ROBVIS-II tool, an advancement refined from its original version to better suit the evaluation of non-randomized intervention studies [66]. The ROBVIS-II tool, known for its comprehensive evaluation of bias risk, was selected due to its suitability for our review’s focus on varied physical activity interventions and their impacts on older adults with neurodegenerative diseases.

An independent examination was conducted for each selected study, thoroughly investigating key aspects such as study design, participant selection and classification, fidelity to intervention protocols, methods of outcome measurement, and the management of potential confounders and missing data. This rigorous analysis was pivotal in determining the methodological strength of the studies and identifying any potential biases.

To guarantee the accuracy and impartiality of our evaluations, we addressed any discrepancies encountered during the assessment process with utmost rigor and transparency. Instances of disagreement or ambiguity regarding the studies’ quality or risk of bias were diligently resolved through a consensus-building approach. This entailed detailed discussions among members of our review team, ensuring a collaborative and consensus-based resolution that led to unanimous agreement on the quality assessments of the included studies.

### 2.5. Data Analysis

In our comprehensive systematic review, which scrutinized the impact of physical activity on the adjunctive management of neurodegenerative diseases among older adults, we employed a sophisticated approach to data analysis that integrated both narrative synthesis and thematic analysis. This blended analytical framework was pivotal in dissecting, interpreting, and amalgamating pivotal findings from interventional studies, thereby providing insightful conclusions on how physical activity influences neurodegenerative disease management in the elderly. Below is an elucidation of how each analytical method contributed to our investigation:Narrative synthesis: Serving as the foundation of our data analysis, narrative synthesis was instrumental in offering a detailed and systematic evaluation of the collected data. This method allowed us to go beyond the simple aggregation of findings, enabling a critical appraisal of interventional studies that explore physical activity’s role in neurodegenerative diseases. Through narrative synthesis, we aimed to distill insights regarding the types, frequency, intensity, and overall efficacy of physical activity interventions [67]. This process involved an in-depth examination of how these interventions could potentially slow disease progression, alleviate symptoms, or enhance the quality of life for older adults with neurodegenerative diseases. The narrative synthesis yielded a comprehensive narrative that identifies trends, evidentiary gaps, and emerging areas for future research within the domain of physical activity and neurodegenerative disease management [68].Thematic analysis: In tandem with narrative synthesis, thematic analysis played a crucial role in pinpointing recurrent themes across the included studies. This qualitative method dove into the specifics of physical activity interventions, identifying core themes such as exercise adherence, motivational factors, intervention scalability, and the role of personalized exercise programs [69]. By focusing on these themes, we delved into the nuances of designing and implementing physical activity programs that are both effective and adaptable to the diverse needs of older adults with neurodegenerative diseases. Thematic analysis illuminated the multifaceted aspects of physical activity interventions, from the perspectives of feasibility and acceptability to the barriers and facilitators influencing their success. This analytical approach enriched our understanding of the intricate dynamics at play in tailoring physical activity as a therapeutic strategy for neurodegenerative diseases, underscoring the critical role of tailored interventions, caregiver support, and interdisciplinary collaboration.

## 3. Results

### 3.1. Risk of Bias

The risk of bias assessment shown in Figure 2 reflects a thorough evaluation of 19 studies included in the systematic review [47,48,49,50,51,52,53,54,55,56,57,58,59,60,61,62,63,64,65]. The overall findings indicate that the vast majority of studies presented a low risk of bias across all of the assessed domains, suggesting a high level of methodological robustness within this body of research.

Specifically, most studies such as those by Rivas-Campo et al., 2023, Pallesen et al., 2019, and Vieira de Moraes Filho et al., 2020, among others, had uniformly low risk ratings in each category—namely, the randomization process, deviation from the intended intervention, missing outcome data, measurement of the outcome, and selection of reported results. This consistency underpins the credibility of their findings and reinforces the overall validity of the conclusions drawn from these studies.

However, Baker et al., 2010 showed a high risk of bias in the domain of deviation from the intended intervention, which is a notable exception in an otherwise low-risk landscape. This particular high risk could affect the reliability of the intervention’s outcome and should be weighed appropriately when considering the study’s contributions to the evidence base. Similarly, while Liu et al., 2022, and Pompeu et al., 2012 exhibited some concerns regarding deviation from the intended interventions, their overall risk of bias remained low, suggesting that these concerns are not pervasive enough to significantly detract from the trustworthiness of the studies’ results.

Noteworthy is the study by Chen et al., 2023, which stands out for its high risk of bias in the measurement of the outcome and some concerns in the selection of reported results. These factors imply potential measurement bias and selective reporting, which may influence the study’s findings and interpretations. Consequently, the results from Chen et al., 2023, should be integrated with caution in the systematic review’s synthesis. In the case of Tyndall et al., 2013, and Maeneja et al., 2023, some concerns were noted in the category of missing outcome data and measurement of the outcome, respectively. While these concerns were not reflected in the overall low risk of bias, they highlight areas where the studies could improve.

In conclusion, the aggregate evidence from the risk of bias assessment points to a predominantly reliable and methodologically sound collection of studies, with isolated issues that warrant careful consideration. The overall low risk of bias across the majority of studies provides a solid foundation for drawing reliable conclusions in the systematic review, while the few instances of higher risk or some concerns emphasize the need for a cautious and balanced interpretation of some individual study outcomes. 

### 3.2. Main Outcomes 

The endeavor to enhance critical care environments for the elderly is an intricate pursuit that has been illuminated by contemporary research. As we navigate through the complexities of aging populations and the escalating demands on healthcare systems, it becomes imperative to understand the multifaceted impact of such enhancements. The main outcomes of recent scholarly investigations revealed four critical themes: the amplification of clinical outcomes, the elevation of patient experience, the diminution of healthcare utilization, and the economic repercussions thereof. Appendix A contains the main extraction information from the included studies. This section endeavors to unravel these threads, presenting a tapestry rich with the potential benefits and inherent challenges of implementing tailored interventions in critical care settings. Drawing on a body of evidence from 19 meticulously selected studies, we aim to showcase the expansive spectrum of benefits that such interventions confer upon the well-being and care efficacy of older adults in high-stakes healthcare environments.

Cognitive function and neurodegenerative disease management: This theme encapsulates studies focusing on the impact of physical activity on cognitive functions in individuals with neurodegenerative conditions. The studies by Rivas-Campo et al., 2023 [50] and Baker et al., 2010 [51], for instance, demonstrated that interventions such as high-intensity intervallic functional training (HIFT) and high-intensity aerobic exercises can improve cognitive assessments scores like the Montreal Cognitive Assessment (MoCA) and various executive function tests. These improvements suggest that physical activity can mitigate some cognitive decline associated with conditions like mild cognitive impairment (MCI) and potentially delay the progression of neurodegenerative diseases. The key takeaway is that targeted physical activities can be a vital part of the adjunctive management of cognitive health in aging populations.Physical function in neurodegenerative conditions: This theme revolves around how physical interventions can affect the physical capabilities of individuals with conditions such as Parkinson’s Disease and mild dementia. Progressive resistance training and neurofunctional training were shown to enhance motor function, reduce symptoms like bradykinesia, and improve gait parameters. The study by Koo, Jang, and Kwon, 2021 [54] highlighted the importance of dual-task training in improving gait, which is crucial for maintaining independence and reducing the risk of falls. These findings are particularly important for developing exercise programs that address not just the cognitive, but also the physical challenges faced by individuals with neurodegenerative diseases.Exercise modalities and adaptation: Under this theme, the focus is on the diversity of exercise modalities and their adaptability to the individual’s needs and conditions. Tai Chi and yoga, as evidenced in the studies by Sungkarat et al., 2016 [66], and Hariprasad et al., 2013 [60], not only support cognitive health, but also offer physical benefits that contribute to lower fall risk and better overall well-being. These modalities are adaptable and can be modified to suit various physical abilities, making them accessible to a broader range of participants. Additionally, the feasibility of adapting exercise programs to online platforms, as explored by Morris et al., 2023 [53], opens up possibilities for remote participation, which is particularly relevant in the context of ongoing concerns about social distancing and accessibility.Physical exercise in terms of intensity and duration and its effects on the body: Aerobic interventions typically involve moderate-to-vigorous intensity activities such as brisk walking or cycling, lasting 30 to 60 min per session, and conducted three to five times weekly. These regimens aim to optimize cardiovascular health and cerebral blood flow, potentially enhancing neuroplasticity. Similarly, resistance training interventions incorporate moderate-to-high intensity exercises targeting major muscle groups, lasting 45 to 60 min, and conducted two to three times weekly. Such programs seek to improve functional capacity, balance, and mobility, crucial for older adults managing neurodegenerative conditions. Mind–body exercises like Tai Chi and yoga, characterized by low-to-moderate intensity movements, mindfulness, and breathing techniques, typically last 60 to 90 min, held two to three times weekly. These modalities emphasize proprioception, balance, and flexibility, offering additional benefits of relaxation and stress reduction. Individualized prescription of exercise intensity and duration, considering baseline fitness, disease severity, and preferences, is paramount for optimizing therapeutic outcomes and enhancing the quality of life in this population.Quality of life and daily functioning: This theme is concerned with the broader impact of physical activity on the quality of life and daily functioning of individuals with neurodegenerative diseases. The studies presented interventions such as dance and Wii-based training, which were found to not only provide physical and cognitive benefits, but also improve symptoms of depression, bodily discomfort, and even the ability to carry out daily activities. For instance, the Dance for PD^®^ model study by Carapellotti et al., 2022 [52] showed potential for emotional, mental, and social health benefits, illustrating how physical activity interventions can transcend physical health and contribute to holistic well-being.Tailored interventions and specific outcomes: Finally, this theme focuses on how personalized interventions cater to specific outcomes and needs. For example, Maeneja et al., 2023 [55] found that aerobic exercise was more effective than cognitive dual-task walking in improving cognitive attention in stroke patients. Similarly, Kim et al., 2016 [56] showed that a combination of physical exercise with a multicomponent cognitive program might be more beneficial for cognitive function than cognitive exercises alone. These findings underscore the importance of designing interventions that are tailored to the specific cognitive and physical needs of the elderly with neurodegenerative diseases, thereby maximizing the benefits of exercise programs.

## 4. Discussion

This systematic review aimed to evaluate the effectiveness of physical activity interventions in the adjunctive management of neurodegenerative diseases among older adults. The 19 studies included in the review encompassed a diverse range of physical activity modalities including aerobic training, resistance training, dance, yoga, dual-task training, and Tai Chi, among others. The findings reveal promising evidence that tailored physical activity programs can benefit older adults with mild cognitive impairment, dementia, Parkinson’s disease, and other neurodegenerative conditions. This discussion integrates the insights gleaned from the reviewed studies to provide an in-depth analysis of the role of physical activity in four critical domains: (1) preserving cognitive function, (2) maintaining physical function and mobility, (3) improving overall well-being and quality of life, and (4) the importance of tailored interventions targeting specific outcomes.

### 4.1. Preserving Cognitive Function

Cognitive decline represents one of the most significant concerns accompanying neurodegenerative diseases in older adulthood [70]. Across several studies, various forms of physical activity demonstrated benefits for cognitive performance in older adults with mild cognitive impairment (MCI) or dementia. For instance, Rivas-Campo et al. (2023) found that high-intensity interval training (HIIT) led to enhancements in global cognition, attention, and verbal fluency in individuals with MCI compared to the controls [47]. The challenge of preserving executive function amid progressive cognitive decline was addressed by Baker et al. (2010), where high-intensity aerobic exercise improved executive control processes in older adults with amnestic MCI, with particularly pronounced benefits for women [48].

These cognitive benefits align with emerging evidence on physical activity’s neuroprotective effects. Animal research indicates that exercise promotes hippocampal neurogenesis, synaptic plasticity, and angiogenesis in the brain [71]. In humans, exercise is associated with increased gray matter volume, white matter integrity, and neural connectivity in regions vital for cognition [72]. The upregulation of neurotrophic factors like brain-derived neurotrophic factor (BDNF) and insulin-like growth factor-1 (IGF-1) is implicated as a key mechanism underlying exercise-induced cognitive improvements [73,74]. 

The control of brain-derived neurotrophic factor (BDNF) levels, an important regulator of neurogenesis and synaptic plasticity, goes beyond just physical activity and involves an intricate combination of genetic, epigenetic, and environmental elements [75]. Recent developments in the fields of neurobiology and exercise physiology have revealed various ways in which BDNF expression is controlled such as through the activation of internal signaling pathways, the adjustment of neurotransmitter systems, and an enhancement of neurovascular coupling [76]. Although studies have shown varying results on the immediate drop in BDNF levels after exercise, the overall increase from regular physical activity highlights the flexible regulation of BDNF [77,78]. Additionally, the growing understanding of different neurotrophins such as GDNF highlights the importance of fully grasping the dynamics of neurotrophins in relation to physical activity for the better management of neurodegenerative diseases like Parkinson’s disease [33,36,79]. 

Research indicates that older adults experience a decrease in brain-derived neurotrophic factor (BDNF) concentrations following activities like Nordic walking and yoga. This reduction might result from the increased use of BDNF for synaptic repair during exercise. Despite this temporary decrease, regular physical activity can actually enhance cognitive function and reduce depressive symptoms in the elderly by boosting the baseline BDNF levels over time. Thus, consistent exercise is beneficial for maintaining brain health and emotional well-being in older populations [80,81].

Exercise-induced muscle metabolism significantly influences the onset of molecular and physiological changes, notably through the secretion of the hormone irisin. This myokine, released during physical activity, is instrumental in mediating several beneficial effects of exercise [82,83]. One of its key roles is serving as a precursor to changes in the concentration of brain-derived neurotrophic factor (BDNF). When muscles contract during exercise, they produce irisin, which then enters the bloodstream and can cross the blood–brain barrier [84]. Once in the brain, irisin stimulates the production of BDNF, a vital factor involved in the growth and maintenance of neuronal cells. BDNF is essential for brain health, supporting neurogenesis, enhancing learning and memory, and maintaining synaptic plasticity [85].

Beyond aerobic and resistance training, mind–body practices like Tai Chi and yoga are gaining momentum as complementary exercise modalities for cognitive health [86]. Sungkarat et al. (2016) demonstrated enhanced memory, executive function, and visuospatial abilities in older adults with MCI following Tai Chi training [87]. A meta-analysis of 18 RCTs corroborated these benefits, underlining Tai Chi’s potential in delaying cognitive decline [88]. Likewise, yoga’s promise was evidenced by Hariprasad et al. (2013), where yoga practice improved memory, attention, executive function, and processing speed in seniors without dementia, surpassing the waitlist controls [89]. Yoga’s neuroprotective effects may arise from its integrated stimulation of physical, cognitive, and emotional well-being [90].

Nordic walking and aqua aerobics offer diverse yet effective exercise options for older adults with neurodegenerative diseases. Nordic walking, with its specially designed poles, provides a low-impact cardiovascular workout that enhances aerobic capacity and balance [91]. Aqua aerobics, conducted in a supportive aquatic environment, improves cardiovascular fitness, muscular strength, and flexibility while reducing the risk of injury [92]. Both modalities offer versatile and enjoyable ways to address the physical and cognitive needs of older adults, making them valuable components of comprehensive intervention programs for neurodegenerative diseases.

Physical activity shows considerable promise in counteracting disease-related cognitive decline. Optimal cognitive gains may necessitate adequately intense aerobic activity [93] coupled with enriched sensorimotor stimulation [94] and mindfulness-based practices [95]. The cognitive heterogeneity of MCI and dementia warrants personalized programs catering to the individual’s baseline impairments [96]. Technology-assisted training is an emerging avenue for adaptable and engaging cognitive exercise [97]. Collaborations between researchers, clinicians, and caregivers will be vital in translating these insights into effective community-based programs for cognitive preservation [98].

### 4.2. Maintaining Physical Function and Mobility

While cognitive health represents a key priority, optimizing physical functioning is equally critical for the autonomy, mobility, and quality of life of older adults amid progressive neurodegeneration [99]. Multiple studies have demonstrated that tailored exercise interventions can improve gait, balance, and mobility in individuals with Parkinson’s disease and dementia. Progressive resistance training enhanced motor symptoms and gait parameters in Parkinson’s patients in research by Vieira de Moraes Filho et al. (2020) and Smaili et al. (2018) [60,61]. Koo, Jang, and Kwon (2021) showed that dual-task gait training significantly improved stride length, velocity, and phase time versus single-task training in older adults with mild dementia, highlighting its potential to bolster mobility [51].

These findings align with the growing consensus on the benefits of exercise for physical functioning in neurodegenerative disease. Resistance training enhances muscle strength and motor control [100], whereas task-specific balance and gait protocols improve stability and mobility in Parkinson’s and dementia patients [101]. Exercise may optimize neuromuscular control mechanisms via improved sensorimotor integration and neuroplasticity [102]. Promoting activity earlier in the disease process could be pivotal for long-term, real-world physical functioning [103].

Considering that the cardiorespiratory fitness level is crucial when designing exercise interventions for older adults managing neurodegenerative diseases [104], it is important to assess the baseline fitness guide tailored programs, ensuring safety and effectiveness. Improving cardiorespiratory fitness correlates with various health benefits, emphasizing its importance in enhancing overall well-being during this stage of life [105].

Furthermore, the feasibility and safety of unsupervised exercise in these patients’ daily lives warrant deeper investigation [106]. Adherence remains a challenge, necessitating creative engagement strategies and telehealth solutions [106]. Caregiver involvement and individually adapted programs are paramount for translating clinical improvements into sustained real-world functioning [106]. Overall, physical activity shows promise in maintaining and improving mobility amid neurodegeneration, but successful implementation demands a multifaceted approach in attending to the patients’ capabilities, environment, support systems, and personal preferences [107].

### 4.3. Improving Overall Well-Being and Quality of Life

While cognitive and physical domains represent key priorities, it is vital to recognize that individuals are greater than the sum of isolated functions. Physical activity provides a holistic opportunity to enhance overall well-being and quality of life for those facing neurodegenerative diseases. Dance interventions, for instance, can integrate motor, cognitive, and psychosocial stimulation. Carapellotti et al. (2022) demonstrated that a 12-week Dance for Parkinson’s program improved symptoms of depression and discomfort while bolstering functional mobility in individuals with Parkinson’s disease [49]. Morris et al. (2023) verified the practicality of transitioning such dance programs online to enhance accessibility [50,108].

Likewise, exergames that interweave physical, cognitive, and social engagement may enrich the quality of life in neurodegenerative disease. Wii Fit training enhanced activities of daily living while providing an enjoyable group activity for Parkinson’s patients [50]. Exergames have shown emotional and social benefits for seniors with MCI, underscoring their multidimensional impact [109]. Further research should investigate their synergistic cognitive and physical effects.

Physical activity also counters the social isolation that frequently accompanies neurocognitive disorders. Group-based programs provide meaningful social connections and peer support that could strengthen cognitive resilience [110]. Taking a holistic perspective, exercise enables individuals to focus beyond disease-defined losses and recognize their sustained capabilities to experience joy, relationships, meaning, and mastery in daily life [111,112].

Therefore, physical activity interventions for neurodegenerative disease should assess not just isolated functions, but multidimensional outcomes like mood, discomfort, activities of daily living, mobility in everyday settings, participation, and subjective well-being [113]. Centering the individual’s priorities and preferences will be key to promoting adherence and long-term gains in quality of life [114,115]. Healthcare policies should provide resources and incentives to implement personalized community exercise initiatives that acknowledge neurodegenerative disease patients as whole people embedded within a social context [116].

### 4.4. Importance of Tailored Interventions Targeting Specific Outcomes

While physical activity confers multidimensional benefits, optimized gains necessitate interventions tailored to the individual’s specific impairments and goals. General physical activity guidelines provide a beneficial starting point but may fail to address the complex, heterogeneous deficits accompanying neurodegenerative diseases [24]. As an illustration, Maeneja et al. (2023) discovered that aerobic training provided greater improvements in cognitive attention compared to dual-task treadmill training in stroke patients with cognitive decline [52]. Stroke survivors commonly experience cognitive challenges impacting daily functioning, warranting interventions to address these impairments. Additionally, the potential overlap in symptoms and therapeutic strategies between stroke-related cognitive impairments and neurodegenerative diseases justifies their inclusion, allowing for a comprehensive exploration of the efficacy of physical interventions across various cognitive and functional domains [117]. Therefore, despite etiological differences including stroke cohorts with cognitive impairments, this enriches our understanding of the broader implications of physical interventions for cognitive health and functional outcomes in older adults [118].

Additionally, multicomponent programs integrating diverse forms of stimulation may confer advantages over single modalities. For instance, Kim et al. (2016) found that the combination of physical exercise with cognitive training enhanced the cognitive function in Alzheimer’s patients beyond gains from cognitive stimulation alone [53]. However, exercise diversity should be balanced with the simplicity of implementation for caregivers [53]. Virtual reality technologies could enable personalized, adaptable multicomponent cognitive challenges [119].

Several notable enhancements are apparent in the reviewed studies when analyzing how various types of physical activities impact cognitive functions in elderly individuals with neurodegenerative disorders. Brisk walking and cycling workouts have been linked to improvements in executive functions like attention, inhibition, and working memory. For example, a meta-analytic review of randomized controlled trials showed notable enhancements in executive functions, as assessed through standardized cognitive assessments, after a 12-week aerobic exercise program [120]. Likewise, resistance training programs have been found to have beneficial impacts on cognitive flexibility and processing speed, leading to faster reaction times and enhanced task-switching abilities in individuals who participate in resistance exercises that focus on major muscle groups [121]. Additionally, Tai Chi and yoga have been associated with improvements in overall cognitive function, especially in areas like verbal memory, visuospatial skills, and processing speed. A study found that older adults with mild cognitive impairment showed marked enhancements in verbal memory after participating in Tai Chi for 6 months, in contrast to a control group receiving the usual care [122]. These results emphasize the various cognitive advantages of various physical activities, underlining the significance of including a variety of methods in comprehensive programs for elderly individuals with neurodegenerative conditions.

Finally, the individual’s priorities and preferences must direct interventions. For example, dance and Tai Chi may suit individuals seeking enriching physical activities with a social component, while home-based resistance training may better fit those prioritizing privacy and convenience [123]. Telehealth solutions could enhance the accessibility of community exercise programs [124]. However, not all individuals feel comfortable with technology, underscoring the need for varied options [125]. Ultimately, carefully matching evidence-based programs to the individual’s goals, environment, and lifestyle is critical to enable long-term adherence and maximize outcomes [126].

Therefore, while general physical activity guidelines provide a valuable starting point, optimal gains for those with neurodegenerative diseases depend on detailed assessments determining the individual’s specific deficits, preferences, and values, followed by tailored programming targeting personalized goals [127]. This requires collaboration among diverse healthcare professionals to integrate medical, psychological, and rehabilitation perspectives [128]. Caregiver education and involvement is equally essential to enable supported implementation in the patients’ daily lives [129]. Further research should continue investigating adapted and integrated modalities spanning recreational, functional, and psychological.

## 5. Conclusions

This systematic review synthesized evidence from 19 studies to evaluate the effectiveness of nursing based physical activity interventions in the adjunctive management of neurodegenerative diseases among older adults. The findings reveal promising support for the benefits of exercise and physical activity in improving cognitive function, physical function, mobility, overall well-being, and quality of life in individuals with mild cognitive impairment, dementia, Parkinson’s disease, and other neurodegenerative conditions. The studies encompassed a diverse range of tailored interventions including aerobic training, resistance training, Tai Chi, yoga, dual-task exercises, dance, and technology-assisted modalities.

Overall, the review indicates that regular physical activity should be a key component of neurodegenerative disease management plans to optimize the outcomes and quality of life. However, several limitations must be considered when interpreting the evidence. Most studies had small sample sizes and relatively short follow-up periods. There was heterogeneity in the populations, interventions, and outcome measures, limiting comparability. Longer RCTs with standardized measures are needed to clarify the optimal physical activity prescriptions. Adherence remains a challenge, and many interventions may be difficult to sustain long-term without adequate support. Additionally, most studies focused on Alzheimer’s and Parkinson’s disease, with few examining other neurodegenerative conditions. Finally, the studies rarely addressed cost-effectiveness or caregiver impacts, which are essential to consider for real-world implementation.

In conclusion, physical activity interventions appear to be beneficial for older adults with neurodegenerative diseases, but higher-quality research is required to strengthen this evidence base. Individualizing activity plans based on the patients’ needs and capabilities is key to enhancing the outcomes. Collaborative approaches engaging clinicians, therapists, caregivers, communities, and policymakers will be vital to translate these findings into feasible, sustainable programs that can optimize functioning and quality of life for a growing population of older adults facing neurodegenerative diseases.

## Figures and Tables

**Figure 1 life-14-00597-f001:**
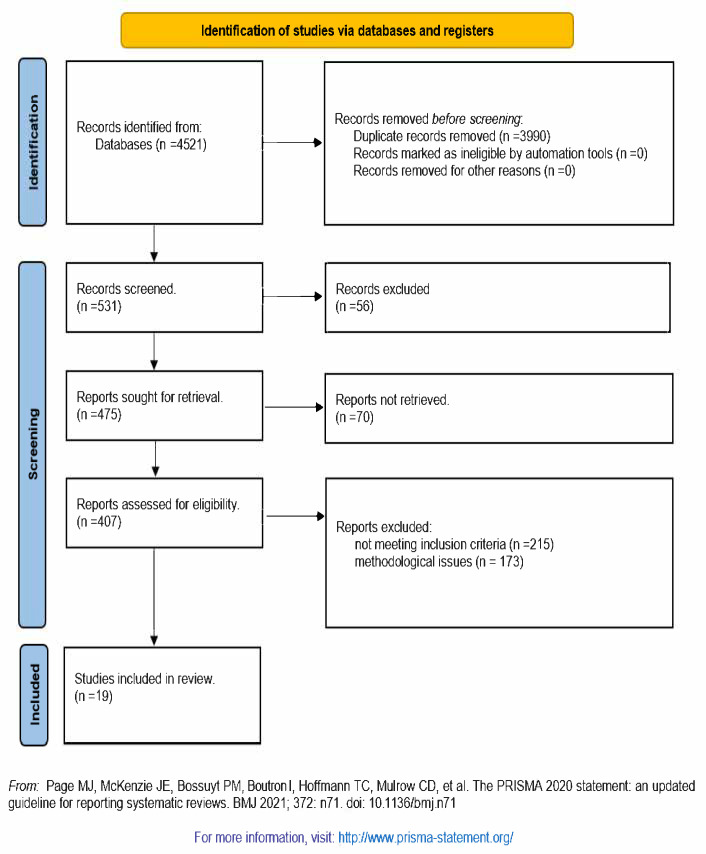
PRISMA flowchart of the included studies.

**Figure 2 life-14-00597-f002:**
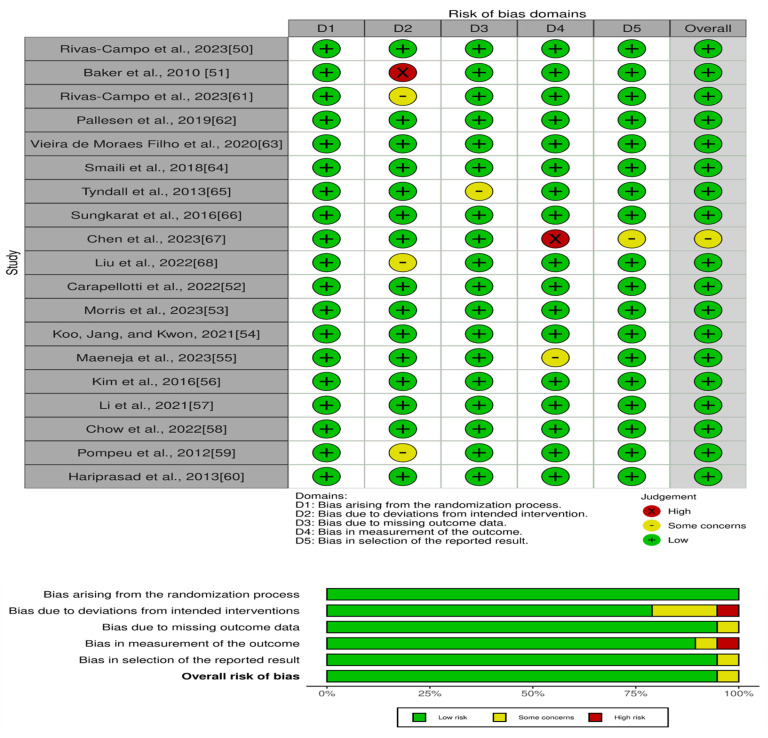
Risk of bias assessment of the included studies.

**Table 1 life-14-00597-t001:** Search strategy.

Database	Search Terms
PubMed	(“Neurodegenerative Diseases”[Mesh] OR “Alzheimer Disease” OR “Parkinson Disease” OR “Dementia”) AND (“Physical Activity”[Mesh] OR “Exercise” OR “Motor Activity”) AND (“Aged”[Mesh] OR “Older Adults”) AND (“Intervention Studies”[Mesh] OR “Randomized Controlled Trials” OR “Clinical Trials”)
MEDLINE	Same as PubMed
Embase	(‘neurodegenerative diseases’/exp OR ‘alzheimer disease’ OR ‘parkinson disease’ OR ‘dementia’) AND (‘physical activity’/exp OR ‘exercise’ OR ‘motor activity’) AND (‘aged’/exp OR ‘older adults’) AND (‘intervention studies’/exp OR ‘randomized controlled trials’ OR ‘clinical trials’)
Web of Science	TS = (neurodegenerative diseases OR alzheimer disease OR parkinson disease OR dementia) AND TS = (physical activity OR exercise OR motor activity) AND TS = (aged OR older adults) AND TS = (intervention studies OR randomized controlled trials OR clinical trials)
Cochrane Library	“Neurodegenerative Diseases” OR “Alzheimer Disease” OR “Parkinson Disease” OR “Dementia” AND “Physical Activity” OR “Exercise” OR “Motor Activity” AND “Aged” OR “Older Adults” AND “Intervention Studies” OR “Randomized Controlled Trials” OR “Clinical Trials”
Google Scholar	(“Neurodegenerative Diseases” OR “Alzheimer Disease” OR “Parkinson Disease” OR “Dementia”) AND (“Physical Activity” OR “Exercise” OR “Motor Activity”) AND (“Aged” OR “Older Adults”) AND (“Intervention Studies” OR “Randomized Controlled Trials” OR “Clinical Trials”)
Scopus	(TITLE-ABS-KEY (neurodegenerative diseases) OR TITLE-ABS-KEY (alzheimer disease) OR TITLE-ABS-KEY (parkinson disease) OR TITLE-ABS-KEY (dementia)) AND (TITLE-ABS-KEY (physical activity) OR TITLE-ABS-KEY (exercise) OR TITLE-ABS-KEY (motor activity)) AND (TITLE-ABS-KEY (aged) OR TITLE-ABS-KEY (older adults)) AND (TITLE-ABS-KEY (intervention studies) OR TITLE-ABS-KEY (randomized controlled trials) OR TITLE-ABS-KEY (clinical trials))

## Data Availability

Data available upon request from the corresponding author.

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
