# Peer review of "The Role of Physical Activity in Adjunctive Nursing Management of Neuro-Degenerative Diseases among Older Adults: A Systematic Review of Interventional Studies"

_life, 2024, doi:10.3390/life14050597_

Round 1

Reviewer 1 Report

Comments and Suggestions for Authors

The manuscript " The Role of Physical Activity in Managing Neurodegenerative Diseases Among Older Adults: A Systematic Review of Interventional Studies" addresses an interesting social issue regarding the importance of physical activity in adjunctive therapy for neurodegenerative diseases. After reading the manuscript, I believe it is a humanistic approach to the problem rather than a clinical analysis taking into account human anatomy, physiology and biochemistry. A positive aspect of the paper is its methodological correctness.

My comments include:

1. the topic of the work: Physical activity cannot be directly indicated as a treatment for neurodegenerative diseases is a false inference. Physical activity can be an adjunctive therapy for the treatment of not only neurodegenerative diseases. such wording occurs throughout the manuscript.

2 The abstract of the paper is too general and the conclusions are not very specific and obvious. No specific therapeutic indications.

Introduction:

1. what is the selection scheme for age-related neurodegenerative diseases? Multiple sclerosis does not affect the elderly as an age-related change.

2. analysis of BDNF neurotrophin levels is very limited. Not BDNF alone is relevant to neurogenesis. Many factors modify this effect. There is a lack of information from the frontiers of neurobiology and exercise physiology. What causes an increase in the production of BDNF? What happens in the muscle during exercise, what effect does BDNF observed in plasma have on its uptake by nerve cells? Physical activity is not the only factor causing an increase in BDNF concentration.

3 Scientific publications indicate a decrease in BDNF after physical training and there is this mechanism explained. The importance of other neurotrophins (GDNF) in Parkinson's disease in particular has also been described. Domaszewska, K., Koper, M., Wochna, K., Czerniak, U., Marciniak, K., Wilski, M., & Bukowska, D. (2020). The effects of Nordic walking with poles with an integrated resistance shock absorber on cognitive abilities and cardiopulmonary efficiency in postmenopausal women. Frontiers in Aging Neuroscience12, 586286.

4 The authors did not accurately characterize physical exercise in terms of intensity and duration and its effects on the body.

5. cognitive functions were mentioned several times in the manuscript. But does any physical activity improve them?

6. In the research methodology, the words used to review the databases are too general

7. The analysis of the study results did not include many physical activities chosen by the elderly, Nordic Walking, Aqua aerobics, why Tai Chi?

8. The discussion did not show the results of studies that did not confirm the purpose of the work, in which the results did not improve any explanation of the mechanisms of occurrence of post-workout adaptive changes.

9. The authors did not refer to the level of physical fitness of this cardiorespiratory, important in this period of life.

Author Response

Dear Reviewer,

Thank you for your thoughtful review of our manuscript, "The Role of Physical Activity in Managing Neurodegenerative Diseases Among Older Adults: A Systematic Review of Interventional Studies." We appreciate your insightful comments and suggestions, which have greatly contributed to the refinement of our work.

Below, we address each of your points and outline the modifications made to the manuscript, highlighted in blue for ease of identification:

  1. We acknowledge your concern regarding the wording throughout the manuscript regarding physical activity as a treatment for neurodegenerative diseases. We have revised the language to reflect that physical activity serves as an adjunctive therapy rather than a direct treatment for these conditions. This modification aims to align more accurately with the current understanding of the role of physical activity in managing neurodegenerative diseases.

  2. We have revised the abstract to provide more specific therapeutic indications and ensure clarity regarding the conclusions drawn from the systematic review. This includes highlighting the specific benefits of physical activity interventions for cognitive function, physical functioning, mobility, daily activities, balance, motor symptoms, and quality of life among older adults with neurodegenerative diseases.

Introduction:

  1. We have clarified the rationale for the selection of age-related neurodegenerative diseases included in the study, emphasizing their prevalence and impact on the elderly population. Additionally, we have addressed the mention of multiple sclerosis in the introduction, providing context for its relevance in the discussion section, particularly within the local context.

  2. We have expanded the analysis of brain-derived neurotrophic factor (BDNF) neurotrophin levels to include a more comprehensive discussion of the factors influencing BDNF production and its effects on neurogenesis. This includes elucidating the mechanisms underlying BDNF regulation, such as its production during exercise and its uptake by nerve cells, to provide a deeper understanding of its role in neurodegenerative diseases.

  3. We have incorporated additional literature discussing the mechanisms underlying changes in BDNF levels following physical training, as well as the importance of other neurotrophins, such as glial cell line-derived neurotrophic factor (GDNF), in neurodegenerative diseases. This broader discussion aims to provide a more comprehensive overview of the neurobiological mechanisms involved in the effects of physical activity on brain health.

  4. We have revised the characterization of physical exercise in terms of intensity and duration to provide more detailed descriptions of exercise protocols from the reviewed studies. This includes discussing the specific intensity and duration of various exercise modalities and their effects on the body, aiming to provide a clearer understanding of the therapeutic benefits of physical activity interventions.

  5. We have addressed the mention of cognitive functions in the manuscript by providing evidence from the reviewed studies demonstrating improvements in cognitive function following different types of physical activities. This includes highlighting specific cognitive domains that have shown improvement, such as executive function, attention, memory, and processing speed, to underscore the cognitive benefits of physical activity interventions.

  6. We have refined the research methodology by providing more specific details on the databases reviewed and the search strategy employed. This includes clarifying the databases searched, the search terms used, and the inclusion/exclusion criteria applied to ensure transparency and reproducibility in the systematic review process.

  7. We have expanded the analysis of the study results to include a broader range of physical activities chosen by the elderly, such as Nordic Walking, Aqua Aerobics, and Tai Chi. This includes discussing the specific effects of these activities on cognitive and physical function, as well as their potential implications for the management of neurodegenerative diseases among older adults.

  8. We have addressed the omission of studies that did not confirm the purpose of the work by discussing the limitations of the systematic review and the implications for future research. This includes acknowledging the need for further investigation into the mechanisms underlying post-workout adaptive changes and the potential factors contributing to variability in study outcomes.

  9. We have incorporated references to the level of physical fitness, particularly cardiorespiratory fitness, among older adults, emphasizing its importance in the management of neurodegenerative diseases. This includes discussing the potential role of physical fitness as a moderator of the effects of physical activity interventions on cognitive and physical function.

We believe that these modifications address your concerns and enhance the clarity, accuracy, and completeness of the manuscript. Thank you once again for your valuable feedback, which has helped strengthen our work.

Reviewer 2 Report

Comments and Suggestions for Authors

The manuscript submitted by the author systematically examines the efficacy of physical interventions in managing neurodegenerative diseases. The author included 19 studies in his review and found a list of physical exercises; from aerobic, resistance, dual-task, Tai Chi, yoga, and dance exercises. These interventions showed benefits for cognition, physical functioning, mobility, daily activities, balance, motor symptoms, and quality of life.

There are major points that I have difficulty accepting and I trust the author can pay attention to:

1)    What is the rationale for defining the type of neurodegenerative diseases being included? In fact, the author only focused on 3 based on the search keywords. However, in the introduction, the author discussed Huntington’s Disease and Multiple Sclerosis. If the choice is due to relevance in the local context where the author lives, perhaps this can be mentioned in the Discussion.

2)    It is a common practice in systematic review to have different examiners in searching and extracting information from the literature. The author mentioned "Independent examination". Who is (are) the persons? Resolution of disagreement during the process needs to have more than 1 person. This seems to be a single-author article, so perhaps acknowledgements will be preferred.

3)    I suggest the author replace the word RCT (Line 12) from the abstract as the review did not only include RCT.

4)    The author wrote “… supplementary Table 1 encapsulates the distilled essence of these themes, each a thread woven into the broader tapestry of age-friendly practices”. What is age-friendly practices? Also, there is no diagram/table illustrating the themes produced by the thematic analysis. The author may want to write/discuss more on this aspect.

5)    What are the five points outlined in Section 3.2 (benefits, main themes or domains) and how are those being conceptualized (from the thematic analysis)? In the Discussion (line 424) the author wrote four domains and they look similar but not the same. I am confused as to what those four are.

6)    Stroke cohort with cognitive impairments was included. However, the author may note that such impairment is a consequence of stroke (vascular injury in nature). Crucially, therefore, it is not strictly neurodegenerative (Ross NS et al, 2022). Can the author justify the inclusion?

7)    More importantly, the roles of physical exercise as management and prevention of neurodegenerative diseases are not new (e.g. Sujkowski, 2022, in which the author cited). Please write the strength of the article that the author proposes. For example, this can be in the form of new themes produced, etc.

Minor: Typoe in Figure 1 (PRISMA) caption, and please include funding source (if any).

Author Response

Dear Reviewer,

Thank you for your thorough review and valuable feedback on our manuscript titled "The Role of Physical Activity in Managing Neurodegenerative Diseases Among Older Adults: A Systematic Review of Interventional Studies." We appreciate the opportunity to address your concerns and make necessary revisions to enhance the quality and clarity of our work. correcrtion was highlighted in green ,  Below are our responses to each of your points:

  1. Regarding the rationale for defining the types of neurodegenerative diseases included, we acknowledge your concern. While our focus was initially on three diseases based on search keywords, we understand the need for clarity on the inclusion criteria. We will explicitly mention in the Discussion section the rationale behind our selection

  2. We appreciate your suggestion regarding the independent examination process. As this is a single-author article,  discrepancies during the process was addressed and resolved by asking colleagues

  3. Thank you for bringing up the discrepancy regarding the term "RCT" in the abstract. We will remove this term to accurately reflect the review's inclusion of various study designs beyond randomized controlled trials.

  4. We understand your query about "age-friendly practices" and the lack of illustration for themes produced by thematic analysis. We will provide a clearer explanation of age-friendly practices 

  5. We will ensure clarity on the six points outlined in Section 3.2 and their conceptualization from the thematic analysis. as regarded to the discussion relevant themes was discussed together 

  6. Regarding the inclusion of stroke cohorts with cognitive impairments, we acknowledge your concern about the vascular nature of such impairment. We will justify this inclusion in the manuscript, considering the overlap in symptoms and therapeutic strategies with neurodegenerative diseases.

  7. We recognize the importance of emphasizing the strengths of our article. We will highlight any new themes produced or unique insights provided, particularly in the management and prevention of neurodegenerative diseases through physical exercise.

Additionally, we will correct the typo in the Figure 1 caption and include the funding source, if applicable, to ensure completeness and accuracy.

Once again, we appreciate your thoughtful review and constructive feedback, which will undoubtedly contribute to the refinement of our manuscript. We look forward to your further evaluation of the revised version.

Reviewer 3 Report

Comments and Suggestions for Authors

This systematic review evaluated the effectiveness of physical activity interventions in the management of neurodegenerative diseases in older adults.

Actually, it is a very interesting article, since it is necessary to check if these therapies or exercises are effective in these populations.

It correctly points out the limitations, they included small samples, heterogeneity and short-term follow-up.

And the conclusions are appreciated, as they report that physical activity interventions appear beneficial for older adults with neurodegenerative diseases, but higher quality research is required to strengthen this evidence base.

The paper is well written, it carries out an adequate process with prism, and the study is of great interest, and the conclusions are very important.

Author Response

Dear Reviewer,

Thank you for your thoughtful and positive feedback on our systematic review evaluating the effectiveness of physical activity interventions in managing neurodegenerative diseases among older adults. We appreciate your recognition of the importance of investigating the efficacy of these therapies in this population.

We acknowledge the limitations highlighted in our study, including the inclusion of small sample sizes, heterogeneity among studies, and short-term follow-up periods. These limitations are crucial to address and underscore the need for higher quality research to strengthen the evidence base for physical activity interventions in older adults with neurodegenerative diseases.

We are pleased that you found our paper to be well-written and that you appreciated our use of the PRISMA process. We agree that rigorous methodology is essential in conducting systematic reviews and are glad that our efforts in this regard have been recognized.

Once again, we appreciate your positive evaluation of our study and your insightful comments. Your feedback will undoubtedly contribute to improving the quality and relevance of our research.

Round 2

Reviewer 1 Report

Comments and Suggestions for Authors

Re-review Manuscripts

I thank the author for correcting the manuscript in the indicated points.

I have a few minor comments:

- The author omitted to cite publications that demonstrated and explained the post-workout decrease in BDNF concentrations in older people training NW, and yoga in conjunction with levels of performance, cognitive function and severity of depressive disorders.

1 Domaszewska, Katarzyna, et al. "The effects of Nordic walking with poles with an integrated resistance shock absorber on cognitive abilities and cardiopulmonary efficiency in postmenopausal women." Frontiers in Aging Neuroscience 12 (2020): 586286.

2. GRONEK, J., & DOMASZEWSKA, K. (2022). Effects of yoga training in post-menopausal women. Trends in Sport Sciences, 29(3).

- please correct the editorial of the manuscript- there is a space missing before the literature citation [...].

- still no explanation of the role of exercise-induced muscle metabolism of the onset and the role of irisin as a precursor of BDNF concentration changes. There is a lack of information and explanation of the causes of the post-workout decline in blood BDNF concentration and the adaptive significance of this process.

Author Response

Dear reviwer

thank you for valuable comments

we add the recommended references with explanation of post-workout decrease in BDNF concentrations in older people training (lines 470-487)

we hope that adds can be satisfied

Reviewer 2 Report

Comments and Suggestions for Authors

The author has to bear in mind that it is difficult to re-read the manuscript while at the same time finding the revision that the author did by purely color-coding. In many instances, it is common practice to state the revised part using Line numbers. For example: "Thank you for pointing this out. We have added ......... (Line xxx)".

Point (1): I do not see where the rationale for only including 3 diseases, instead the authors explained MS further (Line 52, Line 103) without being clear why it is relevant. The literature included in the synthesis did not include studies on MS.

Point (5): Again, the author did not specify how those points (now become six points) were conceptualized. Did those points come from the thematic analysis?

Author Response

Dear reviwers 

thank you for your valuable comments.

point one: we have changed the introduction section based on your comments with adding most possible diseases that affect the elderly (lines 30-52)

point 5 : the points mentioned were based on the thematic analysis extracted from the table, also we add the sixth point based on reviewer 1 advise. 

we hope these modifications are satisfied.